# Regulatory-approved deep learning/machine learning-based medical devices in Japan as of 2020: A systematic review

Nao Aisu[1], Masahiro Miyake[1,2]*, Kohei Takeshita[3], Masato Akiyama[2,4], Ryo Kawasaki[2,5], Kenji Kashiwagi[2,6], Taiji Sakamoto[2,7], Tetsuro Oshika[2,8], Akitaka Tsujikawa[1]

1 Department of Ophthalmology and Visual Sciences, Kyoto University Graduate School of Medicine, Kyoto, Japan, 2 Japanese Society of Artificial Intelligence in Ophthalmology, Tokyo, Japan, 3 Department of Innovation for Medical Information Technology, Jikei University School of Tokyo, Tokyo, Japan, 4 Department of Ocular Pathology and Imaging Science, Graduate School of Medical Sciences, Kyushu University, 5 Artificial Intelligence Center for Medical Research and Application, Osaka University Hospital, 6 Department of Ophthalmology, Faculty of Medicine, University of Yamanashi, 7 Department of Ophthalmology, Kagoshima University Graduate School of Medical and Dental Sciences, 8 Department of Ophthalmology, Faculty of Medicine, University of Tsukuba

* miyakem@kuhp.kyoto-u.ac.jp

**Data Availability Statement:** Data that support the findings of this study are available in the JAAME homepage. https://www.jaame.or.jp/ Paid

## Abstract

Machine learning (ML) and deep learning (DL) are changing the world and reshaping the medical field. Thus, we conducted a systematic review to determine the status of regulatory-approved ML/DL-based medical devices in Japan, a leading stakeholder in international regulatory harmonization. Information about the medical devices were obtained from the Japan Association for the Advancement of Medical Equipment search service. The usage of ML/DL methodology in the medical devices was confirmed using public announcements or by contacting the marketing authorization holders via e-mail when the public announcements were insufficient for confirmation. Among the 114,150 medical devices found, 11 were regulatory-approved ML/DL-based Software as a Medical Device, with 6 products (54.5%) related to radiology and 5 products (45.5%) related to gastroenterology. The domestic ML/DL-based Software as a Medical Device were mostly related to health check-ups, which are common in Japan. Our review can help understanding the global overview that can foster international competitiveness and further tailored advancements.

## Author summary

Artificial Intelligence (AI), Machine learning (ML)/deep learning (DL) is in the early stages of its applications in the medical field. The current study, by investigating the state of regulatory-approved ML/DL-based medical devices in Japan, revealed that the clinical application of AI-based medical devices is closely related to society. It also emphasizes the need to understand the industrial demand, and sociocultural situation of each country, such as the state of health insurance, medical access, and health awareness, for global expansion of ML/DL-based medical devices. The study consists one of the two studies

subscription is required for data usage via the search repository "JAAME search". https://search.jaame.or.jp/jaames/login.php.

**Funding:** This research was supported by AMED under Grant Number JP20vk0124003. This research was also supported by MHLW ICT infrastructure establishment and implementation of artificial intelligence for clinical and medical research program, Grant Number JP20AC0201. The funders had no role in study design, data collection and analysis, decision to publish, or preparation of the manuscript.

**Competing interests:** The authors have declared that no competing interests exist.

that revealed the approval status of AI-related medical devices among the leading members of international regulatory harmonization. Revealing the regulatory status of each country provides a global overview that can foster international competitiveness and further tailored advancements of ML/DL-based medical devices. Considering the current lack of relevant information about such devices, further studies revealing the status of regulatory-approved ML/DL-based medical devices are expected.

## Introduction

The ongoing fourth industrial revolution is mainly driven by the development of information technology, computational power, and deep learning (DL), which is a method of machine learning (ML). The fourth industrial revolution is changing the world in several aspects, including the medical and healthcare fields. Thus, ML/DL-based technologies are expected to transform the healthcare field by providing novel insights from the large amounts of data generated from daily practice [1–5]. This technological application would improve disease detection or diagnostic accuracy as well as aid the discovery of unprecedented observations for personalized healthcare development.

In December 2016, a DL-based algorithm was first introduced in a major medical journal [6]. The algorithm demonstrated comparable accuracy to ophthalmologists in grading diabetic retinopathy from fundus images. In the following year, reports emerged of a dermatologist-level algorithm for classifying skin cancer into benign or malignant [7] as well as a pathologist-level algorithm for detecting a metastatic lesion of breast cancer from hematoxylin and eosin-stained tissue sections of lymph nodes [8]. Consequently, ML/DL attracted considerable attention in the medical field, and more research papers on ML/DL algorithms are being published (**Fig 1**) [9–12]. Unlike the increasing number of ML/DL studies [13], there are few regulator-approved, implemented ML/DL algorithms. For example, a recent report noted that only 27

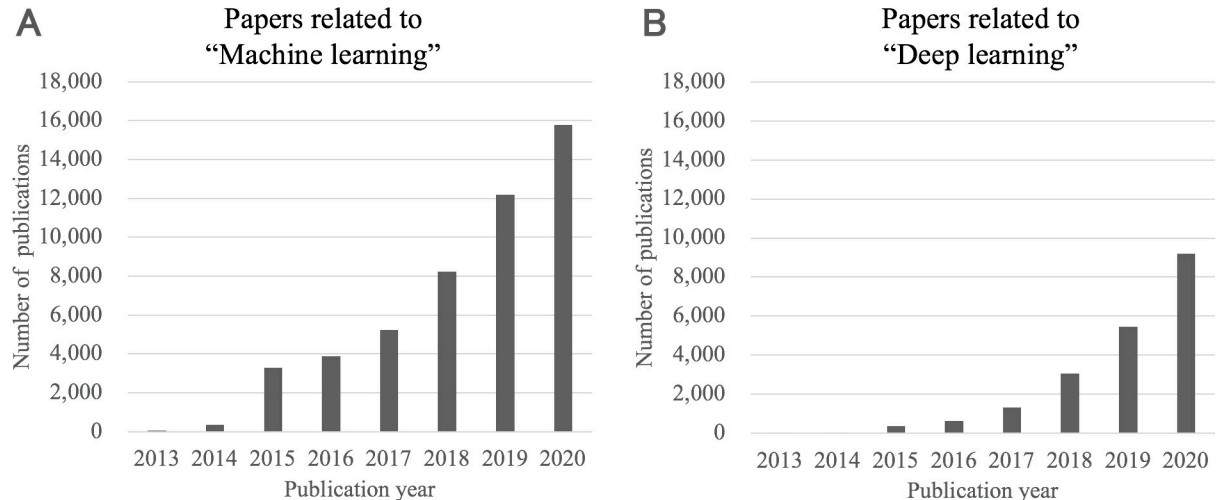

**Fig 1. Transition of the number of publications regarding machine learning or deep learning in PubMed.** (A) Transition of the number of machine-learning-related publications. The search query was (Machine learning) AND (("2015/1/1"[Date—Publication]: "2020/12/31"[Date—Publication])) (accessed at 2, February, 2021). (B) Transition of the number of deep-learning-related publications. The search query was (Deep learning) AND (("2015/1/1"[Date—Publication]: "2020/12/31"[Date—Publication])) (accessed at 2, February, 2021). Publications regarding machine learning or deep learning are rapidly increasing in PubMed.

artificial intelligence (AI)/ML algorithms had been approved by the Food and Drug Administration (FDA) in the United States before February 2020 [14]. Although this discrepancy is due to several reasons, the most prominent is that regulatory authorities require the algorithms to have a high level of reliability in terms of clinical effectiveness and safety; moreover, the regulatory authorities strictly check other factors, such as the reliability of the training data and teacher labels as well as the validity of the validation data and their use cases.

Currently, the regulatory authority of each country independently reviews the application of AI-based medical devices because there are no harmonized regulatory standards. However, harmonization is necessary, considering the growing importance of AI-based medical devices that are utilized across borders. The International Council for Harmonization (ICH) was launched in April 1990 to mobilize regulatory authorities and the pharmaceutical industry for the development of global harmonization in pharmaceutical regulations. The ICH was originally founded by the regulatory agencies and industry associations of Europe, Japan, and the United States. The regulatory bodies of each nationality (the European Medicine Agency (EMA), FDA, and the Japanese Pharmaceuticals and Medical Devices Agency (PMDA)) collaborate as leading members of the ICH, which currently includes 17 Members and 32 Observers worldwide. Harmonization in medical devices regulation is also being discussed in the International Medical Device Regulators Forum (IMDRF), of which Japan is also a founding member. IMDRF is expected to lead the global harmonization of the regulatory standards of AI/ML-based medical devices and ensure that safe, effective, and high-quality devices are developed, registered, and maintained in the most resource-efficient manner.

As this is the dawn of AI-based medical devices, the state of regulatory-approved AI-based medical devices in each country has not been sufficiently described; for instance, only one review article has reported the status of the United States [14]. Therefore, we conducted a systematic review to elucidate the status of regulatory-approved ML/DL-based medical devices in Japan, which is a leading member (alongside the United States and European Union) of international regulatory harmonization.

## Methods

We conducted a systematic review to determine the state of regulatory-approved ML/DL-based algorithms for medical practice in Japan. Hence, we searched the Japan Association for the Advancement of Medical Equipment search service (JAAME Search) to obtain the data. JAAME Search is a paid electronic database for medical devices and consists of 12 databases, including a database for the laws and regulations, notifications, package insert, approval/certification, etc. We used the approval/certification database, which covers almost all medical devices in Japan. As the first ML/DL-based algorithm in Japan is known to have been approved in December 2018, the search focused on all Software as a Medical Device (SaMD) that were certified or approved between 1st December, 2018 and 31st October, 2020 (accessed on 23rd November, 2020). From this database, we obtained information such as generic name, marketing name, classification of device, applicant name, application category, application type, date of approval/certification, and approval/certification number.

Furthermore, we selected SaMD that were approved by the Ministry of Health, Labor and Welfare because all ML/DL-based SaMD require an approval by the Ministry regardless of their classification. This criterion is part of the Japanese regulation for marketing approval of medical devices. In Japan, all Class III and Class IV medical devices and a part of Class II medical devices require approval by the Ministry of Health, Labor and Welfare after a review by the PMDA. Whether a Class II medical device needs an approval or not depends on certification standards: if the Ministry of Health, Labor and Welfare has established certification criteria for

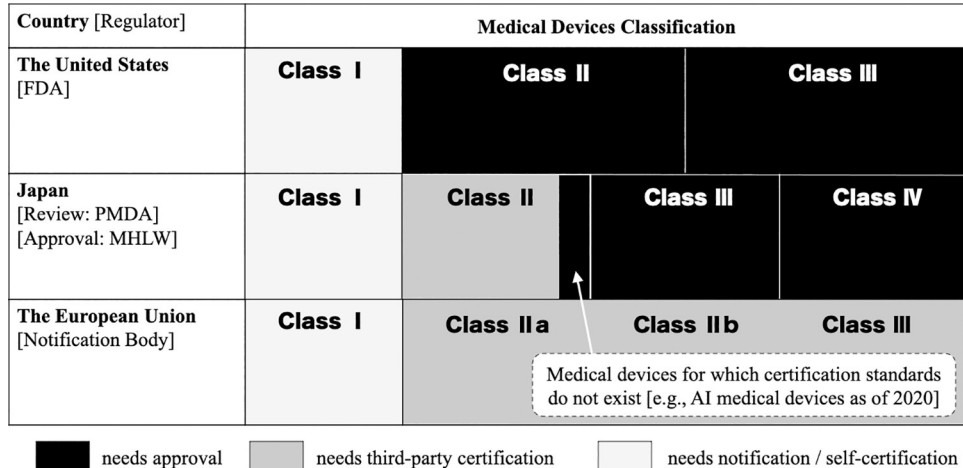

**Fig 2. Comparison of the regulation for marketing approval of medical devices among the United States, Japan, and European Union.** The figure illustrates the regulatory status of the three nationalities based on the classification of medical devices. In Japan, medical devices of class III or more and those of class II for which certification standards do not exist (e.g., artificial intelligence (AI) medical devices as of 2020) require the review of Pharmaceuticals and Medical Devices Agency as well as the approval of the Ministry of Health, Labour, and Welfare. A software that corresponds to class I medical device is not treated as medical device in Japan. FDA: Food and Drug Administration; PMDA: Japanese Pharmaceuticals and Medical Devices Agency; MHLW: Ministry of Health, Labour, and Welfare.

a medical device category, the device does not need an approval; it needs certification instead. Because of the unprecedented nature of ML/DL-based SaMD, ML/DL-based SaMD currently require an approval in Japan. A comparison of the regulations for marketing approval of medical devices among Japan, US, and EU are summarized in **Fig 2**.

The ML/DL-based SaMD were identified by referring to publicly accessible information, including official websites, press-releases, academic papers, and package inserts. If it was clear from the context that an SaMD employed any ML/DL method (e.g., support vector machine (SVM), random forest, convolutional neural network, recurrent neural network, generative adversarial network) within the scope of the approval, we considered the product as an ML/DL-based algorithm. Thus, devices that adopted ML/DL algorithms outside the scope of the approval were not regarded as ML/DL-based algorithms because the ML/DL used in such devices were irrelevant to diagnosis or lesion detection. Regarding devices for which we could not find publicly accessible information that clearly indicates the usage of ML/DL, we contacted the responsible marketing authorization holder. We asked them [1] to provide us the package insert, [2] whether the SaMD employs AI, and [3] to clarify the algorithm employed in the SaMD. If it was still unclear whether the SaMD employed any ML/DL method, we considered the device as a non-ML/DL-based algorithm. Similarly, if the marketing authorization holder refused to provide information or did not respond to our e-mail within two weeks, we considered the device as a non-ML/DL-based algorithm.

Finally, we reviewed each of the identified SaMD and summarized their characteristics.

## Results

The flow diagram of the result counts is presented in **Fig 3**. From the total of 114,150 approvals or certifications of medical devices registered in the JAAME database, we identified 408 approvals or certifications of program devices. One hundred and twenty SaMD (29.4%) were approved or certificated from December 1st, 2018 to October 31st, 2020. Among these, 39 were approved, consisting of 22 (56.4%) Class II SaMD and 17 (43.6%) Class III SaMD. Nine

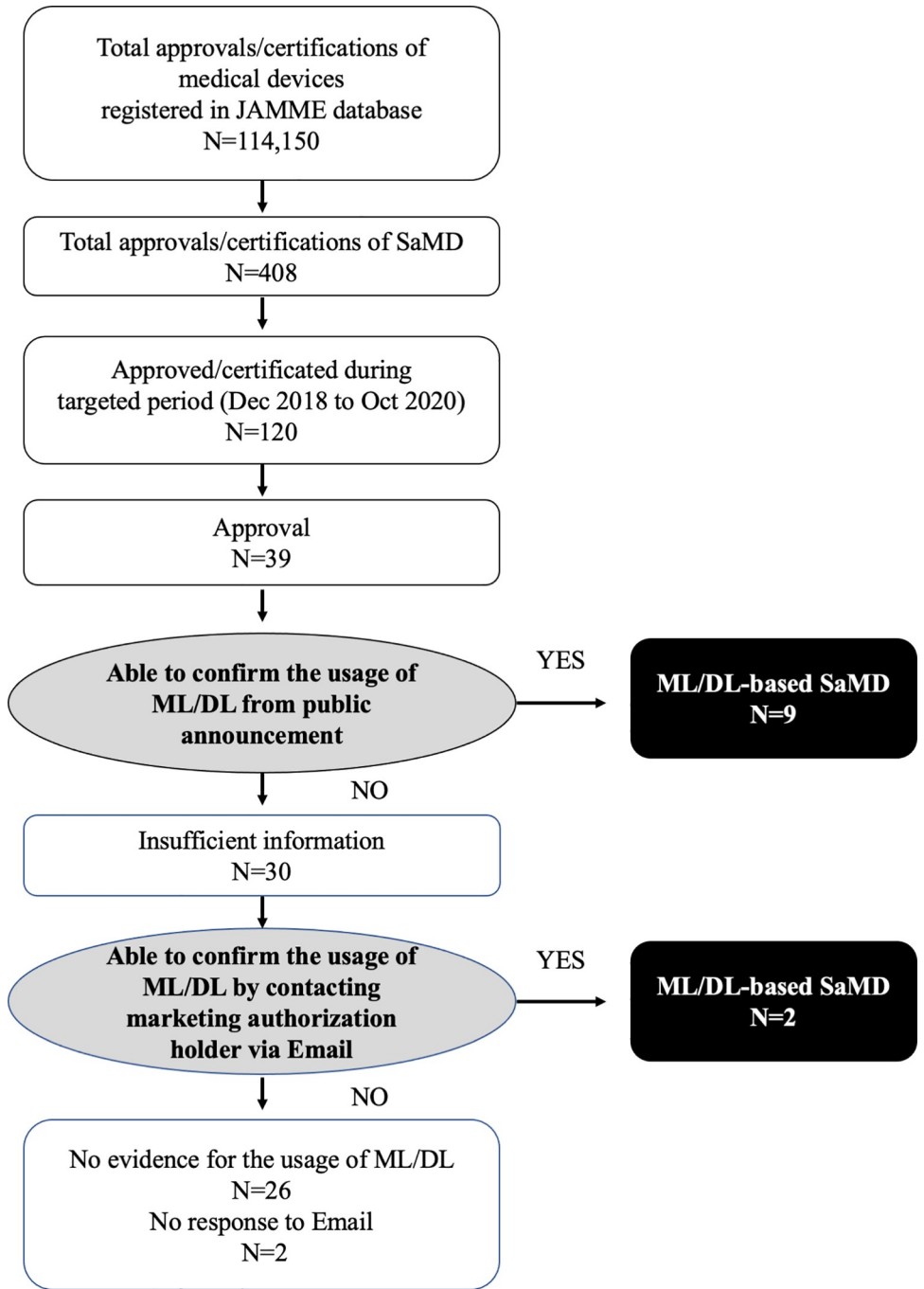

**Fig 3. Flow diagram of identifying the applicable Software as a Medical Device (SaMD).** From the total of 114,150 medical devices in the Japan Association for the Advancement of Medical Equipment (JAAME) search database accessed on 23rd November 2020, we have identified 408 SaMD. Among these, 120 were certificated or approved from December 1st, 2018 to October 31st, 2020 and 39 were approved SaMD. Products with a clear statement of ML/DL usage or with the confirmation from the marketing authorization holder were considered as ML/DL-based. 11 ML/DL-based SaMD were finally identified.

products were confirmed as ML/DL based on publicly available information, and the other 30 products proceeded to direct query for the marketing authorization holders. We received a reply for 28 products, and among them, 2 were confirmed as ML/DL-based. The details of

these two devices for which we did not receive a reply are presented in **S1 Table**. Consequently, 11 ML/DL-based SaMD were finally identified.

An overview of the ML/DL-based medical devices and algorithms is presented in **Table 1**. The first approval was in December 2018, and it was the only ML/DL-based SaMD approved in the year. This was followed by one approval in 2019 and nine more by the end of October 2020. The number of ML/DL-based SaMD approvals in Japan is rapidly increasing, following the trend of the US as shown in **Fig 4**. The figure illustrates a comparison between the current approval number/year in Japan and the US. The status of the US refers to the study conducted by Benjamens et al. [14] The state of 2020 in the US is shown by the simulated number from the study as it terminated in February 2020 unlike our study that terminated in October 2020. Note that the study conducted by Benjamens et al. evaluated the number of regulatory-approved AI/ML-based SaMD, whereas we evaluated regulatory-approved ML/DL-based SaMD. This is because the term "artificial intelligence" is too broad in meaning to represent the recent fourth industrial revolution. Even with discrepancy in terminology between the two studies, the figure illustrates the trend of the rapidly increasing Japanese approval number and the trend that follows the recent status of the US.

As shown in Table **1**, the medical specialties of regulatory-approved ML/DL-based SaMD were biased to two areas: radiology and gastroenterology, with 6 products (54.5%) and 5 products (45.5%) respectively. All algorithms were "locked" algorithms, i.e., an algorithm that provides the same result each time with the same input and does not change or adapt through real-world practice. Three (27.3%) of the devices were programs developed in foreign companies and imported; the domestic marketing authorization holder applied for approval. The other eight (72.7%) devices were domestically developed. All 11 devices were developed for analyzing medical images, such as X-ray, computed tomography (CT), MRI, and endoscopy. When categorized by their function, three (27.3%) were computer-aided diagnosis (CADx) (EndoBrain, EndoBrain-UC, EndoBrain-Plus), seven (63.6%) were computer-aided detection (CADe) (EIRL aneurysm, FS-AI688, InferRead CT Pneumonia, AI-Rad Companion, EndoBrain-EYE, Ali-M3, EIRL X-Ray Lung nodule), and the remaining one (9%) combined both functions of CADx and CADe (EW10-EC02).

The first regulatory-approved AI/ML SaMD in Japan was EndoBrain, which supports the endoscopists' real-time diagnosis of "neoplastic polyps" or "non-neoplastic polyps" during colonoscopy check-ups. By analyzing the images of the lesion taken with an ultra-magnification colonoscope using SVM algorithm, the possibility of being neoplastic or non-neoplastic is displayed as a percentage. The sensitivity of the device was reportedly 96.8%, and its accuracy was 98.0% [15, 16]. EIRL aneurysm is a CADe software for detecting brain aneurysms (diameter ≧ 2 mm) from brain MRA images. This software can support the diagnosis of unruptured cerebral aneurysms at brain check-ups by increasing the sensitivity to detect aneurysms from 68.2% to 77.2% [17]. EndoBrain-UC is a software for predicting the presence or absence of intestinal inflammation (active phase or remission phase) from the ultra-magnified colonoscope images of patients with ulcerative colitis. It could find the remission phase with a sensitivity of 89.5%, specificity of 99.1%, and accuracy of 92%. FS-AI688 is a CADe software for supporting lung nodule detection from lung CT images of suspected lung cancer patients, and it displays marks on possible lung nodules. Both InferRead CT Pneumonia and Ali-M3 are DL-based CADe programs for detecting COVID-19 pneumonia from chest CT images in three probability grades; the devices were developed in China. AI-Rad Companion is a series of imaging supportive software, of which "the Chest CT image contract service" is the only product approved in Japan. This software can detect and quantify abnormalities of multiple areas of the body, such as the lungs, heart, and aorta, from chest CT images. EndoBrain-EYE is a polyp detection program for regular colonoscopy check-ups. When the device detects a

**Table 1. Overview of the ML/DL-based medical devices and algorithms approved in Japan.**

| No. | Device name | Short description | Class | Company name | Application distinction | Approval date | PMDA approval number | Medical specialty | Algorithm | CAD | Country | Image type |
|---|---|---|---|---|---|---|---|---|---|---|---|
| 1 | EW10-EC02 | Lesion detection and diagnosis during colonoscopy | III | FUJIFILM | approved | 2-Sep-20 | 30200BZX00288000 | Gastroenterology | deep learning | CADe +CADx | Japan | endoscopy |
| 2 | EIRL X-Ray Lung nodule | Lung cancer detection | II | LPIXEL | approved | 20-Aug-20 | 30200BZX00269000 | Radiology | deep learning (CNN) | CADe | Japan | X-ray |
| 3 | EndoBRAIN-Plus | Pathological prediction during colonoscopy | III | CYBERNET | approved | 15-Jul-20 | 30200BZX00235000 | Gastroenterology | machine learning (SVM) | CADx | Japan | endoscopy |
| 4 | Ali-M3 | COVID-19 pneumonia detection | II | MIC Medical Corp. | approved | 29-Jun-20 | 30200BZX00212000 | Radiology | deep learning | CADe | China | CT |
| 5 | EndoBRAIN-EYE | Polyp detection during colonoscopy | II | CYBERNET | approved | 29-Jun-20 | 30200BZX00208000 | Gastroenterology | deep learning | CADe | Japan | endoscopy |
| 6 | AI-Rad companion (Chest CT) | Detects and quantifies abnormalities of the chest CT | II | Siemens Healthineers Japan | approved | 19-Jun-20 | 30200BZX00202000 | Radiology | deep learning | CADe | Germany US | CT |
| 7 | InferRead CT Pneumonia | COVID-19 pneumonia detection | II | CES decartes | approved | 3-Jun-20 | 30200BZX00184000 | Radiology | deep learning (CNN) | CADe | China | CT |
| 8 | FS-AI688 | Lung cancer detection | II | FUJIFILM | approved | 8-May-20 | 30200BZX00150000 | Radiology | deep learning | CADe | Japan | CT |
| 9 | EndoBRAIN-UC | Inflammation assessment support for Ulcerative Colitis | III | CYBERNET | approved | 27-Apr-20 | 30200BZX00136000 | Gastroenterology | machine learning (SVM) | CADx | Japan | endoscopy |
| 10 | EIRL aneurysm | Cerebral aneurysm detection | II | LPIXEL | approved | 17-Sep-19 | 30100BZX00142000 | Radiology | deep learning (CNN) | CADe | Japan | MRI |
| 11 | EndoBRAIN | Pathological prediction during colonoscopy | III | CYBERNET | approved | 6-Dec-18 | 23000BZX00372000 | Gastroenterology | machine learning (SVM) | CADx | Japan | endoscopy |

CNN: convolutional neural network, SVM: support vector machine, CADe: computer-aided detection, CADx: computer-aided diagnosis; PMDA: Pharmaceuticals and Medical Devices Agency (Japan)

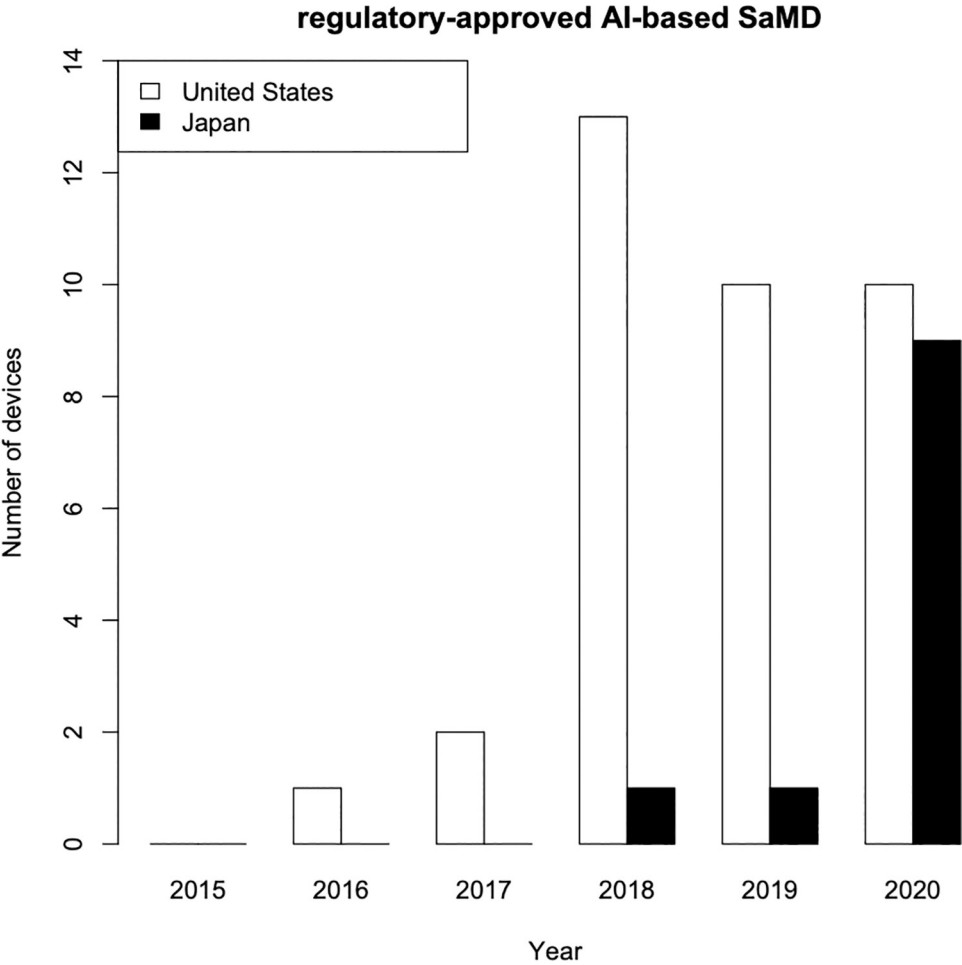

**Fig 4. Comparison of the approval status of machine learning/deep learning-based software as a medical device (SaMD) between Japan and the United States.** The figure illustrates a comparison between the current approval number/year in Japan and the US. The status of the US refers to the study conducted by Benjamens et al. [14] The state of 2020 in the US is shown by the simulated number from the study as it terminated in February 2020 unlike our study that terminated in October 2020. Note that the study conducted by Benjamens et al. evaluated the number of regulatory-approved AI/ML-based SaMD, whereas we evaluated regulatory-approved ML/DL-based SaMD. This is because the term "artificial intelligence" is too broad in meaning to represent the recent fourth industrial revolution. Even with discrepancy in terminology between the two studies, the figure illustrates the trend of the rapidly increasing Japanese approval number and the trend that follows the recent status of the US.

candidate lesion for polyp or cancer in the real-time colonoscopy images, it alerts the colonoscopists with sounds and colors on the screen. A sensitivity of 95% and specificity of 89% were reported for the device [18]. EndoBrain-Plus analyzes images of the lesion acquired using an ultra-magnifying colonoscope under methylene blue staining during colonoscopy check-ups. The system displays in real-time the probability of the lesion to be "non-neoplastic," "adenoma," "intramucosal carcinoma," or "invasive carcinoma." Its performance to detect invasive carcinoma is reported to be 91.8% of sensitivity and 97.3% of specificity. EIRL X-Ray Lung nodule detects suspected pulmonary nodules (5 mm to 30 mm) on chest x-ray images. This CADe is reported to increase a sensitivity to detect lung nodules by 9.9 percentage points for radiologists and 13.1 percentage points for non-specialists. EW10-EC02 is a software for colonoscopy, which supports lesion detection, such as polyps, by framing the suspected lesion on

the monitor and alerting with sounds. The device can also differentiate the lesion as neoplastic or non-neoplastic. These functionalities of the device operate in real-time during colonoscopy.

## Discussion

The implementation of ML/DL in medical devices is advancing swiftly as ML/DL demonstrates high prospects in improving clinical outcomes and public health, and it is cost-effective [19, 20]. In these early stages of the ML/DL era, we aimed to highlight the state of regulatory-approved ML/DL-based SaMD in Japan whose regulatory body, PMDA, is one of the leading members of international regulatory harmonization alongside Europe's EMA and the United States' FDA. Our study showed that 11 AI/ML-based SaMD had been approved in Japan as of October 2020.

In comparison to the US approval status reported by Benjamens, et.al [14]., the field of radiology is a trendsetter of ML/DL-based SaMD in both countries as SaMD related to radiology was the most approved both in Japan and the US with 54.5% and 72.4%, respectively. This is probably due to the image-intensive nature of radiology, and DL technology has high affinity to imaging. Conversely, a prominent feature of the Japanese approval status is that gastroenterology was second (45.5%) to radiology in the implementation of ML/DL in the medical field, whereas the FDA-approved devices next to radiology were in cardiology, internal medicine/endocrinology, neurology, ophthalmology, emergency medicine, and oncology. The Endo-Brain series occupies four of five regulatory-approved endoscopy-related ML/DL-based SaMD, which were developed with funding from the Japan Agency for Medical Research and Development (AMED), technical support from Cybernet, and endoscopic technology and commercialization from Olympus. Olympus is a Japanese company occupying 70% of the world market share for endoscopes. The technology to enhance image resolution with ultra-magnification colonoscopes and their powerful sales channel is believed to have been a major driving force behind the implementation of ML/DL technology in this field. Another unique device among the domestic devices is EIRL aneurysm, an algorithm to detect "unruptured" aneurisms from MRA images. Although several algorithms have been developed in other countries that support the detection of acute lesion, such as cerebral injury or hemorrhage using MRI, EIRL aneurysm is unique because it focuses on the precursor lesion placing MRI as a health check-up tool. This was approved relatively early, meaning the demand for the product was high. As Japan is a global leader in MRI possession, "Brain check-up" is becoming more popular among Japanese citizens because of its accessibility and affordability. This situation likely facilitated the implementation of this technology.

The first autonomous diagnostic device approved by the FDA (April 2018) was a diabetic retinopathy screening device (IDx-DR, Digital Diagnostics) that was recently followed by a similar device approved in August 2020 (EyeArt, EYENUK). In contrast, there have been no regulatory-approved ML/DL-based SaMD in the field of ophthalmology in Japan. This difference may be related to the social need as Japan has easier medical access with the availability of several medical practitioners in local clinics as well as universal health coverage. Nevertheless, as the Japanese Ocular Imaging registry was established as a platform for industry-university collaboration, similar to the Intelligent Research in Sight registry of the American Academy of Ophthalmology, the number of ML/DL-based ophthalmological device will hopefully increase. Building a platform for AI utilization is an important issue in every country, and Japan is moving toward the realization. In addition, to support a rapid implementation of AI-based SaMD, the Ministry of Health Labor and Welfare and PMDA is taking a strategic approach to promote early approval of SaMD. This package is named the Digital Transformation Action Strategies in Healthcare (DASH) for SaMD. More details of DASH for SaMD are provided in **S1 Note**.

Apparently, AI-based SaMD are closely related to society. Thus, focusing on the public needs or industrial strengths of each country can lead to a more competitive and unique development when seeking a global market. For example, health check-ups are popular in Japan because of the increasing public health awareness and easy access to medicine. Most ML/DL-based SaMD approved in Japan were programs that targeted check-ups rather than a specialized or specific medical practice. Moreover, the high proportion of ML/DL-based SaMD related to endoscopy in Japan reflects Japan's industrial strength in endoscopic systems (**S1 Fig**). [21] It is important for each country to scrutinize the public demand and industrial situation to facilitate the implementation of AI. Knowledge of the developmental background is also crucial when considering the implementation of a foreign device as most ML/DL-based medical devices will probably be exported after domestic commercialization. Imported products may not work well due to differences in backgrounds (e.g., social needs, health insurance system, etc.), even if the product is dominant in other countries. Therefore, the approval situation and background information of the innovation can be an important information that suggest how to adopt imported ML/DL-based devices. Therefore, it is worthwhile to investigate the developmental situation of SaMD in each country.

In this systematic review, we report on the regulatory-approved ML/DL-based SaMD in Japan. Our review helps understanding the overview of the global developmental situation of ML/DL-based medical devices, considering the current lack of relevant information about such devices. As ML/DL-based SaMD are closely related to the industrial and cultural backgrounds of each country, development that is focused on the strengths of each country can lead to international competitiveness. Additionally, the maximization of the capabilities of AI requires a consideration of the strengths in each country and creation of a platform for more active developments.

## Supporting information

**S1 Table. Summary of the ML/DL-based medical devices without a response from the marketing authorization holders.**
(DOCX)

**S1 Note. Digital Transformation Action Strategies in Healthcare for software as a medical device.** Introduction of the national strategy taken in Japan for earlier promotion and approval of software as a medical device.
(DOCX)

**S1 Fig. Global market share of 6 major medical devices field (2014).** The figure shows that the Japanese companies dominate the global market share of the endoscopic market.
(DOCX)

## Acknowledgments

We would like to thank the marketing authorization holders for collaborating by answering our queries on the early stage of this work.

## Author Contributions

**Conceptualization:** Nao Aisu, Masahiro Miyake.

**Data curation:** Nao Aisu, Masahiro Miyake, Kohei Takeshita.

**Formal analysis:** Nao Aisu, Masahiro Miyake.

**Funding acquisition:** Masahiro Miyake.

**Investigation:** Nao Aisu, Kohei Takeshita.

**Methodology:** Nao Aisu, Masahiro Miyake.

**Project administration:** Nao Aisu, Masahiro Miyake.

**Resources:** Kohei Takeshita.

**Software:** Nao Aisu.

**Supervision:** Masahiro Miyake, Masato Akiyama, Ryo Kawasaki, Kenji Kashiwagi, Taiji Sakamoto, Tetsuro Oshika, Akitaka Tsujikawa.

**Validation:** Masahiro Miyake, Masato Akiyama, Ryo Kawasaki, Kenji Kashiwagi, Taiji Sakamoto, Tetsuro Oshika, Akitaka Tsujikawa.

**Visualization:** Nao Aisu, Masahiro Miyake.

**Writing – original draft:** Nao Aisu.

**Writing – review & editing:** Nao Aisu, Masahiro Miyake, Kohei Takeshita, Masato Akiyama, Ryo Kawasaki, Kenji Kashiwagi, Taiji Sakamoto, Tetsuro Oshika, Akitaka Tsujikawa.

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
