## [Decision Letter · Decision Letter 0]

30 Jul 2021

PDIG-D-21-00007

Regulatory-approved Deep Learning/Machine Learning-Based Medical Devices in Japan as of 2020: A Systematic Review

PLOS Digital Health

Dear Dr. Miyake,

Thank you for submitting your manuscript to PLOS Digital Health. After careful consideration, we feel that it has merit but does not fully meet PLOS Digital Health’s publication criteria as it currently stands. Therefore, we invite you to submit a revised version of the manuscript that addresses the points raised during the review process.

As per comments from the reviewer, please address the concerns and resubmit to us again for consideration.

We look forward to receiving your revised manuscript.

Kind regards,

Matthew Chua Chin Heng

Academic Editor

PLOS Digital Health

Journal Requirements:

1. We do not publish any copyright or trademark symbols that usually accompany proprietary names, eg (R), (C), or TM  (e.g. next to drug or reagent names). Therefore please remove all instances of trademark/copyright symbols throughout the text.

2. Please provide a detailed Financial Disclosure statement. This is published with the article, therefore should be completed in full sentences and contain the exact wording you wish to be published.

i). State what role the funders took in the study. If the funders had no role in your study, please state: “The funders had no role in study design, data collection and analysis, decision to publish, or preparation of the manuscript.”

ii). If any authors received a salary from any of your funders, please state which authors and which funders.

3. In the online submission form, you indicated that your data will be submitted to a repository upon acceptance.  We strongly recommend all authors deposit their data before acceptance, as the process can be lengthy and hold up publication timelines. Please note that, though access restrictions are acceptable now, your entire data will need to be made freely accessible if your manuscript is accepted for publication. This policy applies to all data except where public deposition would breach compliance with the protocol approved by your research ethics board. If you are unable to adhere to our open data policy, please kindly revise your statement to explain your reasoning and we will seek the editor's input on an exemption. Please be assured that, once you have provided your new statement, the assessment of your exemption will not hold up the peer review process.

Additional Editor Comments (if provided):

Dear Authors

Thank you for your submission to PLOS Digital Health. As per comments from the reviewer, please address the concerns and resubmit to us again for consideration.

Reviewers' comments:

Reviewer's Responses to Questions

**Comments to the Author**

1. Does this manuscript meet PLOS Digital Health’s publication criteria? Is the manuscript technically sound, and do the data support the conclusions? The manuscript must describe methodologically and ethically rigorous research with conclusions that are appropriately drawn based on the data presented.

Reviewer #1: Partly

2. Has the statistical analysis been performed appropriately and rigorously?

Reviewer #1: N/A

3. Have the authors made all data underlying the findings in their manuscript fully available (please refer to the Data Availability Statement at the start of the manuscript PDF file)?

Reviewer #1: Yes

4. Is the manuscript presented in an intelligible fashion and written in standard English?

Reviewer #1: Yes

5. Review Comments to the Author

Reviewer #1: This paper is a systematic review article that summarizes ML/DL in medical devices in Japan. Considering Japan's position in ICH, the summary of regulation-applied M/DL-based software as a medical device in Japan has a significant meaning in digital health.

My comments for the revision of the paper are as follows.

1. It points out errors in the methodology. The authors say that if the marketing authorization holder refuses to provide information or does not respond to an email, this classification has an error. It means just the company did not provide the data from the authors' requests.

Even if the company did not provide information, the authors should add more information about these technologies by utilizing public data registered in the regulation authorization such as medical specialty, country, image type, etc. can be obtained from public data.

If this information is not added, this study lists only 11 technologies obtained from six companies and cannot be considered a review of Japan's ML/DL-based SaMD. Therefore, it is very limited information, and it has low research value.

2. If this study is about regulation-approximated ML/DL-based SaMD in Japan, we would like a brief description of the approval process of SaMD in Japan PMDA and similarities/differences in FDA.

3. Further investigation is needed to determine how much ML/DL-based SaMD is currently being used within the Japanese healthcare market. Discussion said that the application of relatively more ML/DL-based SaMD to the endoscope field than the U.S. reflects the Japanese medical market. However, it is also necessary to provide evidence for actual sales and use.

6. PLOS authors have the option to publish the peer review history of their article (what does this mean?). If published, this will include your full peer review and any attached files.

**Do you want your identity to be public for this peer review?** For information about this choice, including consent withdrawal, please see our Privacy Policy.

Reviewer #1: No

---

## [Editor Report · Decision Letter 1]

27 Oct 2021

Regulatory-approved Deep Learning/Machine Learning-Based Medical Devices in Japan as of 2020: A Systematic Review

PDIG-D-21-00007R1

Dear Dr. Miyake,

We're pleased to inform you that your manuscript has been judged scientifically suitable for publication and will be formally accepted for publication once it meets all outstanding technical requirements.

Within one week, you'll receive an e-mail detailing the required amendments. When these have been addressed, you'll receive a formal acceptance letter and your manuscript will be scheduled for publication.

An invoice for payment will follow shortly after the formal acceptance. To ensure an efficient process, please log into Editorial Manager at https://www.editorialmanager.com/pdig/ click the 'Update My Information' link at the top of the page, and double check that your user information is up-to-date. If you have any billing related questions, please contact our Author Billing department directly at authorbilling@plos.org.

Kind regards,

Matthew Chua Chin Heng

Academic Editor

PLOS Digital Health